# Eupatilin Ameliorates Hepatic Fibrosis and Hepatic Stellate Cell Activation by Suppressing β-catenin/PAI-1 Pathway

**DOI:** 10.3390/ijms24065933

**Published:** 2023-03-21

**Authors:** Jinyuan Hu, Yuanyuan Liu, Zheng Pan, Xuekuan Huang, Jianwei Wang, Wenfu Cao, Zhiwei Chen

**Affiliations:** 1Chongqing Key Laboratory of Traditional Chinese Medicine for Prevention and Cure of Metabolic Diseases, College of Traditional Chinese Medicine, Chongqing Medical University, Chongqing 400016, China; 2Institute of Life Sciences, Chongqing Medical University, Chongqing 400016, China; 3Department of Traditional Chinese Medicine, Chongqing College of Traditional Chinese Medicine, Chongqing 402760, China

**Keywords:** eupatilin, hepatic fibrosis, HSC, EMT, β-catenin, PAI-1

## Abstract

The activation of hepatic stellate cells (HSCs) has proved to be pivotal in hepatic fibrosis. Therefore, the suppression of HSC activation is an effective anti-fibrotic strategy. Although studies have indicated that eupatilin, a bioactive flavone found in *Artemisia argyi*, has anti-fibrotic properties, the effect of eupatilin on hepatic fibrosis is currently unclear. In this study, we used the human hepatic stellate cell line LX-2 and the classical CCl4-induced hepatic fibrosis mouse model for in vitro and vivo experiments. We found that eupatilin significantly repressed the levels of the fibrotic markers COL1α1 and α-SMA, as well as other collagens in LX-2 cells. Meanwhile, eupatilin markedly inhibited LX-2 cell proliferation, as verified by the reduced cell viability and down-regulation of c-Myc, cyclinB1, cyclinD1, and CDK6. Additionally, eupatilin decreased the level of PAI-1 in a dose-dependent manner, and knockdown of *PAI-1* using *PAI-1*-specific shRNA significantly suppressed the levels of COL1α1, α-SMA, and the epithelial–mesenchymal transition (EMT) marker N-cadherin in LX-2 cells. Western blotting indicated that eupatilin reduced the protein level of β-catenin and its nuclear translocation, while the transcript level of β-catenin was not affected in LX-2 cells. Furthermore, analysis of histopathological changes in the liver and markers of liver function and fibrosis revealed that hepatic fibrosis in CCl4-treated mice was markedly alleviated by eupatilin. In conclusion, eupatilin ameliorates hepatic fibrosis and hepatic stellate cell activation by suppressing the β-catenin/PAI-1 pathway.

## 1. Introduction

Hepatic fibrosis, a common pathological condition of chronic liver disease, is characterised by the excessive accumulation of the extracellular matrix (ECM) [1]. According to recent estimates, approximately 1–2% of people suffer from hepatic fibrosis worldwide, and more than one million people die from hepatic fibrosis every year [2]. Although it is reversible [3], if left untreated, hepatic fibrosis may further develop into cirrhosis, and even liver cancer [4]. In a healthy liver, hepatic stellate cells (HSCs) are vitamin A-storing cells in a quiescent state [5]. However, liver injury induces quiescent HSCs into an activated state. Activated HSCs are the main source of ECM components and α-smooth muscle actin (α-SMA), which play a significant role in the development of hepatic fibrosis [6]. Therefore, one of the most effective strategies for hepatic fibrosis treatment is to inhibit the activation of HSCs [7,8].

Increasing evidence shows that the Wnt/β-catenin signalling pathway plays a vital role in the prevention and treatment of hepatic fibrosis, and that the activation of β-catenin can promote ECM accumulation and the proliferation of HSCs [9]. Plasminogen activator inhibitor-1 (PAI-1), a member of the serine protease inhibitor family, is a downstream target gene of the Wnt/β-catenin signalling pathway. The siRNA-mediated inhibition of PAI-1 can effectively suppress hepatic fibrosis [10], while PAI-1 overexpression promotes ECM deposition and organ fibrosis [11]. PAI-1 has emerged as a promising novel target for hepatic fibrosis treatment [12]. Therefore, there is tremendous value in screening the inhibitor of PAI-1. Previous studies have demonstrated that epithelial–mesenchymal transition (EMT) is involved in hepatic fibrosis progression by promoting the conversion of activated HSCs into myofibroblasts that secrete excessive ECM [13]. HSCs undergo a transdifferentiation to the mesenchymal phenotype during fibrogenesis. This conversion is considered similar to EMT. Therefore, activated HSCs possess the characteristics of cells in the EMT process. PAI-1 has been reported to participate in EMT in a variety of cells [14,15]. However, the correlation between PAI-1 and EMT in HSCs has not been fully elucidated.

Eupatilin is a bioactive flavone identified in *Artemisia argyi* which has been used as a traditional Chinese medicine for thousands of years. Eupatilin displays various pharmacological activities, such as antioxidant, anti-inflammatory, and anti-allergic abilities [16,17,18,19]. Recent evidence suggests that eupatilin relieves TGF-β-induced fibrotic effects in Ishikawa cells and vocal fold fibroblasts [20,21]. In addition, eupatilin effectively ameliorated idiopathic pulmonary fibrosis in vivo [22]. However, the anti-fibrotic effects and underlying molecular mechanisms of eupatilin in hepatic fibrosis remain unclear.

The purpose of this study was to determine the anti-hepatic fibrosis activity of eupatilin and explore its potential molecular mechanisms. We found that eupatilin ameliorated hepatic fibrosis and HSC activation by inhibiting the β-catenin/PAI-1 pathway. Our results indicate a promising natural compound for anti-hepatic fibrosis therapy and an effective PAI-1 inhibitor.

## 2. Results

### 2.1. Eupatilin Represses the Activation of HSCs

The activation of HSCs is a key contributor to hepatic fibrosis. After activation, α-SMA expression is increased, and ECMs, including collagens, are synthesised excessively in HSCs [23]. LX-2 is a cell line derived from activated human HSCs, retaining the key features of activated HSCs. Therefore, LX-2 cells were used in this study to investigate the effect of eupatilin on HSC activation (Figure 1A). As shown in Figure 1B,C, LX-2 cells were treated with different concentrations of eupatilin, and the expression of COL1α1 and α-SMA was down-regulated in a dose-dependent manner. Compared with the control group, the expression of some collagens was suppressed in eupatilin-treated LX-2 cell group, as indicated using RNA sequencing analysis (Figure 1D). In addition, transcriptome analysis revealed that the top five reactome pathways of the differentially expressed genes (DEGs) were enriched in ECM-related pathways, including ECM organization, assembly of collagen fibrils and other multimeric structures, degradation of the ECM, and collagen degradation (Figure 1E). In summary, these results indicate that eupatilin represses the activation of HSCs by reducing the expression of collagens and α-SMA.

### 2.2. Eupatilin Inhibits the Proliferation of HSCs

To accurately reflect the effect of eupatilin on HSC proliferation, CCK-8 and EdU assays were performed in LX-2 cells. The CCK-8 assay is based on measuring the dehydrogenase activity of proliferating cells that are metabolically active and able to transform the slight yellow WST-8 into orange formazan. The degree of the orange was directly proportional to the number of viable cells. The result of the CCK8 assay showed a decreased number of cells as the concentration of eupatilin increased, and suggested eupatilin inhibited the LX-2 cell viability (Figure 2A). EdU (5-ethynyl-2′-deoxyuridine) is nucleoside analog of thymidine and it can incorporate into DNA during DNA synthesis in proliferating cells. As shown in Figure 2B,C the percentage of EdU-labeled cells were significantly decreased under eupatilin treatment, especially in 40 μM and 80 μM groups. The data indicated that eupatilin inhibited HSC proliferation. To further investigate the mechanism, we performed gene set enrichment analysis (GSEA) using transcriptome data. The results showed that seven gene sets decreased significantly in eupatilin-treated group (*p* < 0.05), and four of them are closely associated with cell proliferation, including G2M checkpoint, E2F targets, MYC targets, and mitotic spindle (Figure 2D). Subsequently, we selected the key regulator of the G2M checkpoint (Cyclin B1), c-Myc, and two other crucial regulators of cycle progression (Cyclin D1 and CDK6) to assess the effects of eupatilin on the cell cycle. Western blot analysis verified that CyclinB1, CyclinD1, CDK6, and c-Myc protein levels were markedly decreased as the concentration of eupatilin gradually increased (Figure 2E,F). Overall, the above data demonstrated that eupatilin repressed the proliferation of HSCs.

### 2.3. Eupatilin Inhibits the EMT of HSCs

Previous studies demonstrated that EMT was closely associated with hepatic fibrosis, and that inhibited the EMT process contributes to the suppression of HSC activation [13]. Surprisingly, the GSEA results of our data showed EMT gene set of was significantly down-regulated in the eupatilin-treated group (Figure 3A). Therefore, we speculated that eupatilin suppressed the activation of HSCs partially by inhibiting the EMT process. Subsequently, the expression of the EMT marker N-cadherin was detected to verify our speculation using Western blotting and immunofluorescence. The results showed that eupatilin attenuated N-cadherin expression in a dose-dependent manner (Figure 3B–D), indicating the inhibition of EMT in the eupatilin-treated LX-2 cells. These results implied that eupatilin alleviated the progression of hepatic fibrosis by inhibiting the EMT of HSCs.

### 2.4. PAI-1 Regulates the EMT and Activation of HSCs

To uncover the molecular mechanisms of eupatilin-induced EMT inhibition, the intersection between EMT-related genes and DEGs in the transcriptome data was analysed. Among these common genes, PAI-1 attracted our attention. Previous studies have shown that PAI-1 is strongly associated with EMT and fibrosis [14,24]. Excessive PAI-1 level inhibits the activation of the ECM degradation enzymes, thereby weakening ECM degradation [25]. Nevertheless, after treatment with eupatilin, the relative mRNA level of PAI-1 was dramatically down-regulated in a dose-dependent manner (Figure 4A), and the same was true for the protein expression of PAI-1 (Figure 4B,C). To further investigate the function of PAI-1 in EMT in LX-2 cells, we constructed shRNAs targeting *PAI-1*. As expected, the protein level of PAI-1 was significantly decreased by shRNA1 or shRNA2 compared with that in the negative control group (Figure 4D,E). Additionally, the protein levels of N-cadherin, COL1α1, and α-SMA were impaired by PAI-1 shRNAs (Figure 4D,E). CCK-8 analysis further indicated that the knockdown of *PAI-1* reduced LX-2 cell viability (Figure 4F). In summary, these findings suggest that eupatilin can modulate EMT and HSC activation by inhibiting PAI-1 expression.

### 2.5. The β-Catenin Pathway Is Necessary for Eupatilin-Induced EMT Repression in HSCs

It is widely known that β-catenin is the crucial transcriptional regulator of PAI-1 [26], and stabilized β-catenin can enter the nucleus and activate the transcription of target genes via binding the TCF/LEF family proteins. In addition, the β-catenin signalling pathway plays a crucial role in EMT [27]. As a result, qRT-PCR and Western blotting were used to detect the expression of β-catenin in eupatilin-treated HSCs. As shown in Figure 5A, eupatilin did not affect the mRNA expression of β-catenin. However, as the concentration of eupatilin increased, β-catenin protein expression decreased (Figure 5B,C). Next, we detected the β-catenin protein level in the nucleus and the cytoplasm. We observed that the levels of β-catenin protein in both the nucleus and cytoplasm decreased (Figure 5D,E). In summary, these results suggest that eupatilin can inactivate the β-catenin signalling pathway to slow down the EMT progression of HSCs in hepatic fibrosis.

### 2.6. Eupatilin Alleviates Hepatic Fibrosis in CCl4-Treated Mouse Model

In this study, we detected the anti-fibrotic effect of eupatilin in vivo using a classical CCl4-induced hepatic fibrosis mouse model. The treatment protocol is shown in Figure 6A. We evaluated the changes in cells and collagen in liver tissues from different angles, using H&E, Masson, and Sirius red staining (Figure 6B). H&E staining showed that hepatocytes were arranged radially around the central vein in the normal liver tissue section, and that the hepatic lobule structure was intact. In the model group, the hepatocytes showed vacuolar changes and a disordered arrangement, and the structure of the hepatic lobules was disarranged. Masson and Sirius red staining showed a small number of collagen fibres around the vessel wall in the control group. However, many collagen fibres were accumulated in the liver tissue and extended outwards to the portal area in the model group. As shown in Figure 6B, all three staining results indicated that, compared with the model group, the morphological changes in liver cells and collagen fibre deposition in the eupatilin treatment group gradually improved and tended toward the normal group. The ratio of liver weight to body weight was used to evaluate liver injury. When compared with control group, the ratios were significantly increased in the model group. In addition, the ratios were decreased in eupatilin-treated groups in a dose-dependent manner (Figure 6C). In addition, eupatilin significantly decreased the collagen volume fraction (Figure 6D). The serum marker of liver function was detected, and as shown in Figure 6E, the level of AST was markedly lower in mice treated with eupatilin compared to the model group mice, suggesting that eupatilin promoted the improvement of hepatic function in the mouse. Furthermore, the ECM components COL1α1 and fibronectin were overexpressed in the CCl4-treated model group, and their expression was notably decreased in eupatilin-treated groups (Figure 6F,G). 

In the pharmacotherapy group, we selected the high-dose group (40 mg/kg/day) for subsequent Western blot analysis. The results indicated that the levels of α-SMA, COL1α1, PAI-1, N-cadherin, and β-catenin proteins in the high-dose group were significantly reduced compared to those in the model group (Figure 7A,B). Collectively, these results confirmed that eupatilin can effectively ameliorate hepatic fibrosis in vivo.

## 3. Discussion

During chronic liver injury, the most characteristic alteration of HSCs is transdifferentiation from an inactivated phenotype to a proliferative, myofibroblast-like cell type [28]. This conversion process is termed ‘activation’. Activated HSCs are a central driver of hepatic fibrogenesis. According to our studies and those of other groups, inhibiting HSC activation is an effective therapeutic strategy for hepatic fibrosis [29]. Although substantial research has been performed in recent years on the molecular mechanisms of hepatic fibrosis, there are still no therapeutic drugs available in the clinic [30]. Therefore, effective and safe therapeutic agents are urgently required. Eupatilin is a natural flavonoid isolated from *Artemisia argyi*. Previous studies have indicated that eupatilin exerts an anti-fibrogenic effect on TGFβ1-induced endometrial fibrosis and bleomycin-mediated lung fibrosis [20,22]. Although a mouse HSC line was used to investigate the potential deactivating capacity of eupatilin [22], more studies are needed to verify the anti-hepatic fibrosis effect of eupatilin and elucidate its underlying mechanism. The data presented here provide evidence that eupatilin significantly alleviates hepatic fibrosis and HSC activation by inhibiting the β-catenin/PAI-1 pathway. 

The Wnt/β-catenin signalling pathway plays an important role in organ fibrosis, such as renal fibrosis, lung fibrosis, and hepatic fibrosis [31,32]. Studies have shown that β-catenin is overexpressed in liver tissues with hepatic fibrosis, whereas blocking the Wnt/β-catenin signalling pathway represses HSC activation [33]. Interestingly, our data indicated that eupatilin decreased the protein level of β-catenin in LX-2 cells, but did not influence its transcription. The balance in β-catenin levels is highly regulated by complex-mediated ubiquitination, phosphorylation, and proteasomal degradation [34]. Therefore, the results of our study indicated that eupatilin may promote β-catenin degradation. In addition, eupatilin may take part in the translational suppression of mRNA of β-catenin. Generally, the increased nuclear translocation of β-catenin can activate downstream target genes [35], such as *cyclinD1*, *c-Myc*, and *PAI-1*. The present evidence confirms that eupatilin repressed the accumulation of β-catenin in the nucleus and inhibited the expression of *cyclinD1*, *c-Myc*, and *PAI-1*. Meanwhile, the gene sets of G2M checkpoint, E2F targets, MYC targets, and mitotic spindle were down-regulated in eupatilin-treated LX-2 cells. In addition, CCK-8 and EdU staining assays also showed that eupatilin suppressed cell proliferation in a dose-dependent manner. In conclusion, our data showed that the β-catenin signalling pathway is responsible for eupatilin-induced cell cycle arrest. 

PAI-1 is not only a target gene of the β-catenin signalling pathway [26], but also a major inhibitor of the fibrinolytic system [36]. It is well known that the PAI-1 level is significantly increased in fibrotic tissues, which weakens collagen proteolytic activities and leads to reduced ECM degradation and tissue fibrogenesis [24]. In the present study, PAI-1 levels were markedly reduced in eupatilin-treated LX-2 cells. However, *PAI-1* knockdown effectively decreased the levels of COL1α1 and α-SMA. Thus, our data provide strong evidence that PAI-1 is responsible for the modulation of HSC activation. Recent research has reported that PAI-1 is closely involved in regulating the EMT process in various cell types [14,15]. EMT is defined as the transition of epithelial cells into mesenchymal cells [37]. It is universally accepted that HSCs undergo a transdifferentiation from non-activated vitamin A-storing cells to myofibroblast-like cells during fibrogenesis [38]. Myofibroblast-like HSCs are the main source of ECM substances that accelerate the progress of hepatic fibrosis [39]. It is difficult to reverse activated HSCs into quiescent HSCs, but the inhibition of EMT may provide an available strategy for HSC deactivation. N-cadherin has been identified as a mesenchymal marker of the EMT process. The results from our study confirmed that N-cadherin is down-regulated in eupatilin-treated LX-2 cells. RNA interference of *PAI-1* caused further suppression of N-cadherin expression. Therefore, it can be concluded that eupatilin prevents the EMT process of HSCs via inhibiting PAI-1 expression.

Numerous studies using models of lung, liver, and kidney fibrosis indicated that inhibition of PAI-1 activity or *PAI-1* knockout alleviates fibrosis [40,41,42]. The present study provided an effective anti-fibrosis drug by inhibiting PAI-1 expression. However, *PAI-1* deficiency presented an opposite effect on cardiac fibrosis [43]. Therefore, caution needs to be taken with the treatment of multiple organ fibrosis model by inhibiting PAI-1 expression though eupatilin. In this case, targeted delivery of eupatilin to liver tissue is worthwhile to explore.

Furthermore, we established a CCl4-induced hepatic fibrosis mouse model to demonstrate the anti-hepatic fibrosis effect of eupatilin. Additionally, we explored the underlying mechanism of eupatilin-induced HSC deactivation. Through a series of in vivo and in vitro experiments, it can be concluded that eupatilin significantly relieved hepatic fibrosis and HSC activation by inhibiting the EMT process and proliferation of HSCs via down-regulating the mRNA levels of downstream targets of the β-catenin pathway, including *PAI-1*, *cyclinD1*, and *c-Myc* (Figure 8). Our study provides novel insights into the potential of eupatilin as an anti-liver fibrosis agent.

## 4. Materials and Methods

### 4.1. Reagents and Antibodies

Eupatilin was obtained from Selleck (Shanghai, China). Dimethyl sulfoxide (DMSO) was purchased from Solarbio (Beijing, China). The aspartate aminotransferase (AST/GOT) kit was purchased from the Nanjing Jiancheng Bioengineering Institute (Nanjing, China). The antibodies used were: COL1α1 (A1352, ABclonal, Wuhan, China), α-SMA (MAB1420, R&D Systems, Minneapolis, MN, USA), Cyclin B1 (#12231, Cell Signaling Technologies, Beverly, MA, USA), Cyclin D1 (ab134175, Abcam, Cambridge, MA, USA), CDK6 (#3136, Cell Signaling Technologies), PAI-1 (ab222754, Abcam), β-catenin (ab32572, Abcam), N-cadherin (A19083, ABclonal, Wuhan, China), GAPDH (E021060-03, EarthOx, Millbrae, CA, USA), HRP-goat anti-rabbit lgG (E030120-02, EarthOx, Millbrae, CA, USA), and HRP-goat anti-mouse lgG (GB23301, Wuhan Goodbio Technology, Wuhan, China).

### 4.2. Cell Culture

LX-2, an immortalised human hepatic stellate cell line, was purchased from Guangzhou Cellcook Biotech Co., Ltd. (Guangzhou, China). LX-2 cells were cultured in high-glucose DMEM with 10% foetal bovine serum and 1% penicillin-streptomycin solution, and maintained in a 37 °C and 5% CO_2_ incubator. The cells were passaged regularly every two days.

### 4.3. Cell Viability

The Cell Counting Kit-8 (CCK-8) (Dojindo, Kumamoto, Japan) was used to assess the viability of LX-2 cells. Briefly, LX-2 cells were seeded into 96-well plates at a density of 2400 cells/well. The cells were treated with different concentrations of eupatilin (0, 20, 40, and 80 μM). Approximately 48 h after drug treatment, 10 µL of CCK-8 reagent was added to each well, followed by incubation for 4 h in the cell incubator. The absorbance of each well at a wavelength of 450 nm was measured using a microplate reader (BioTek, Winooski, VT, USA). Three replicates were performed for each group.

### 4.4. EdU Assay

The effect of eupatilin on LX-2 cell proliferation was assessed using the BeyoClick™ EdU Cell Proliferation Kit (Beyotime, Shanghai, China). An appropriate number of LX-2 cells were seeded in six-well plates overnight and treated with different concentrations of eupatilin for 48 h. Then, according to the manufacturer’s instructions, 10 µM of 5-ethynyl-2′-deoxyuridine (EdU) reagent was added to each well and incubated for 2 h. The cells were immobilised with 4% paraformaldehyde at 20–25 °C for 10–15 min, washed, and permeated several times. After cell nuclei were stained with DAPI, fluorescence detection and photograph collection were performed using a fluorescence microscope (Olympus IX73, Tokyo, Japan), and counting analysis was performed using Adobe Photoshop CS5 software (Adobe Systems Inc., San Jose, CA, USA).

### 4.5. RNA Isolation and Real-Time PCR

The eupatilin-treated cells were collected, and the appropriate amount of Trizol and chloroform were added, violently shaken, and then centrifuged (12,000× *g*, 10 min). The supernatant was collected in a centrifugal tube, and the same volume of isopropyl alcohol was added, mixed thoroughly, and centrifuged again. The liquid was drained from the tube and centrifuged with 75% ethanol for further washing. Next, DEPC water was added to adjust and test sample concentrations. PCR amplification conditions were as follows: pre-denaturation at 98 °C for 30 s, denaturation at 98 °C for 5 s, and annealing at 60 °C for 30 s, for a total of 35 cycles and ending with the melt curve procedure. *GAPDH* was used as a control for mRNA expression analysis. The specific primer sequences are listed in Table 1.

### 4.6. Western Blotting

Relative protein level was analysed by Western blotting. Cellular protein extraction was performed using an ice-cold RIPA lysis buffer containing protease inhibitors. Lysates were centrifuged (14,000× *g*, 10 min, 4 °C). Protein concentration was quantified using a bicinchoninic acid kit (Beyotime, Shanghai, China). Equal amounts of protein samples were added to each well and separated using 10% sodium dodecyl sulphate-polyacrylamide gel electrophoresis. The proteins were transferred onto PVDF membranes. The membranes were immediately placed in TBST with 5% non-fat powdered milk and incubated for 2 h at 20–25 °C. Next, the membranes and primary antibodies were incubated at 4 °C for 12–16 h, washed several times with TBST, and incubated with HRP-labelled secondary antibodies for 1–2 h at 20–25 °C. Proteins were labelled using a hypersensitive luminescence solution, and the grey value of the blots was calculated and quantified using ImageJ software according to the reference bands of GAPDH. GAPDH was used as an internal reference to normalize expression levels between samples and avoid the influence of the cell proliferation dose on the expression levels of proteins.

### 4.7. RNA Sequencing Analysis

LX-2 cells were treated with eupatilin (80 µM) for 48 h (n = 3). Total RNA was collected using RNAiso Plus (TaKaRa, Dalian, China), and a NanoDrop 2000 spectrophotometer was used to determine the quality and concentration of total RNA. The Illumina NovaSeq6000 (Illumina, San Diego, CA, USA) platform was used to perform transcriptome sequencing. Clean reads were obtained by removing linkers and low-quality sequences from the raw sequence data. Clean reads were mapped to the *Homo sapiens* (human) reference genome in orientation mode using HISAT2 software (Version 2.1.0, https://daehwankimlab.github.io/hisat2/, accessed on 10 September 2021). The differential expression of gene analysis was screened using DESeq2 R software (Version 1.24.0, Bioconductor, http://bioconductor.org/packages/stats/bioc/DESeq2/, accessed on 10 September 2021) with the following criteria: FDR < 0.05 and FC > 2. Gene set enrichment analysis (GSEA) was performed using the Majorbio platform (Version 3.0, Majorbio, https://cloud.majorbio.com, accessed on 14 November 2021) based on the hallmark gene sets of the Molecular Signatures Database (MSigDB, Version 6.2, UC San Diego and Broad Institute, http://www.gsea-msigdb.org/gsea/login.jsp, accessed on 14 November 2021).

### 4.8. shRNA Interference

Short hairpin RNA targeting PAI-1 (sh-PAI-1) was designed by Tsingke (Beijing, China). The two shRNAs were constructed using the pTSB-SH-copGFP-2A-Puro vector with the sequences ‘CCGG-GCTATGGGATTCAAGATTGAT-CTCGAG-ATCAATCTTGAATCCCATAGC-TTTTTT’ and ‘CCGG-AGACCAACAAGTTCAACTATA-CTCGAG-TATAGTTGAACTTGTTGGTCT-TTTTTT’. These shRNAs were introduced into cells using a Polyethylenimine Linear (PEI) MW40000 reagent (YEASEN, Shanghai, China) for transient transfection.

### 4.9. Animal Procedures and Treatments

In total, 50 c57BL/6J male mice were purchased from the Experimental Animal Centre of Chongqing Medical University. The first week involved adaptive feeding. In the second week, all mice were randomly divided into two groups: a normal group (n = 10) and a CCl4-treated group (n = 40). All mice, except the normal group mice, were intraperitoneally injected with a 20% carbon tetrachloride (CCl4) and corn oil mixture (CCl4:corn oil = 1:4) twice a week for eight weeks. At the end of the fourth week of injection, the CCl4-treated group was again randomly divided into the model group and low-, medium-, and high-dose treatment groups with the intragastric administration of eupatilin at 10, 20, and 40 mg/kg/day, respectively, with 10 mice in each group. Simultaneously, the normal and model groups received 0.5% carboxymethyl cellulose sodium solution. The duration of eupatilin treatment was four weeks. In the ninth week of the animal experiment, the mice were sacrificed by deep anaesthesia and a high concentration of CO_2_ asphyxia method, and the livers were weighed. Throughout the experiment, mice ate and drank freely.

### 4.10. Liver Histopathology

Approximately 1 cm-wide tissue was collected from the maximum leaf edge on the left side of the liver tissue; 1/3 of it was cut and fixed in a 4% paraformaldehyde solution for 48 h, then removed, dehydrated, embedded in paraffin, and sectioned (4 μm). Liver tissue was stained with Hematoxylin and eosin (H&E), Masson’s trichrome, and Sirius red. Stained sections were analysed and calculated using a light microscope and ImageJ software (Version: 1.44, National Institutes of Health, Bethesda, MD, USA).

### 4.11. Statistical Analysis

The data obtained from all experiments were analysed using SPSS 25.0. The images in this study were created using GraphPad Prism 8.0.2 software (GraphPad Prism Inc., La Jolla, CA, USA). Statistical differences between the groups were analysed using a one-way ANOVA analysis of Tukey’s test. *p* < 0.05 indicates statistical differences.

## Figures and Tables

**Figure 1 ijms-24-05933-f001:**
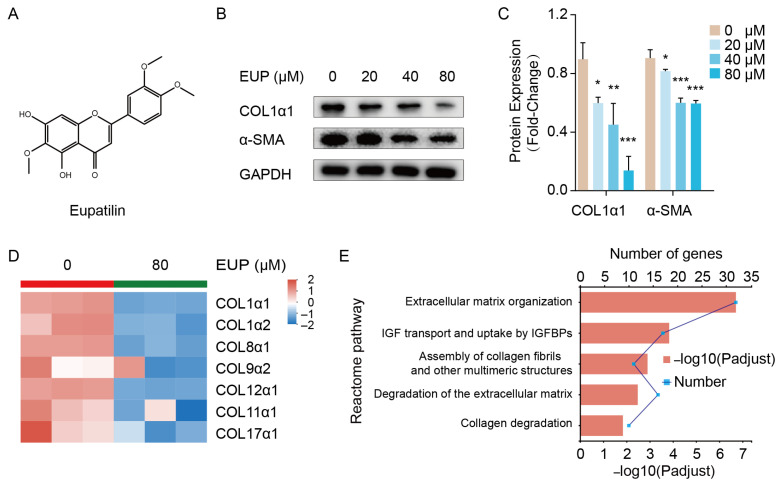
Eupatilin represses the activation of LX-2 cells. (**A**) Chemical structure of eupatilin. (**B**) Protein expression of COL1α1, α-SMA detected 48 h after eupatilin treatment by Western blotting. (**C**) Relative protein signal intensity was quantified as mean ± standard deviation (mean ± SD, n = 3). *p* values are calculated by one-way ANOVA followed by the Tukey’s test. * *p* < 0.05, ** *p* < 0.01 and *** *p* < 0.001 vs. 0 μM EUP group. (**D**) Heatmap showing expression level of collagens evaluated by transcriptome analysis (n = 3). (**E**) The top five reactome pathways of the DEGs in transcriptome analysis. EUP: Eupatilin; α-SMA: α-smooth muscle actin; COL1α1: Collagen type I alpha 1; COL1α2: Collagen type I alpha 2; COL8α1: Collagen type VIII alpha 1; COL9α2: Collagen type IX alpha 2; COL12α1: Collagen type XII alpha 1; COL11α1: Collagen type XI alpha 1; COL17α1: Collagen type X VII alpha 1; GAPDH: Glyceraldehyde-3-phosphate dehydrogenase.

**Figure 2 ijms-24-05933-f002:**
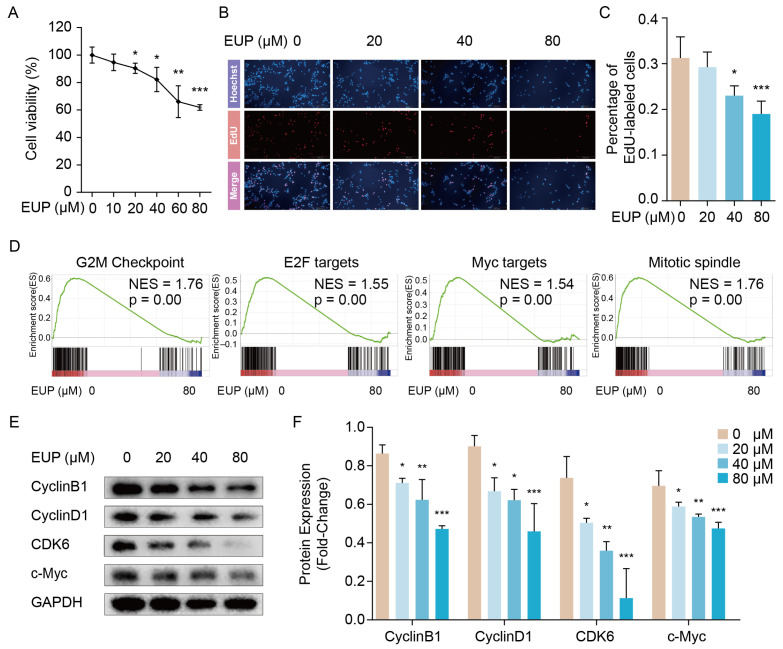
Eupatilin inhibits the proliferation of LX-2 cells. (**A**) Cell viability of LX-2 cells treated with different eupatilin concentrations. (**B**) The effect of eupatilin on DNA replication of LX-2 cells investigated by EdU staining. (**C**) The percentage of EdU-labelled cells is plotted (mean ± SD, n = 4). (**D**) GSEA enrichment plots showed four proliferation-related gene sets were down-regulated in eupatilin exposed LX-2 cells. (**E**) The protein levels of CyclinB1, CyclinD1, CDK6, and c-Myc detected 48 h after eupatilin treatment using Western blotting. (**F**) Relative protein signal intensity was quantified as mean ± standard deviation (mean ± SD, n = 3). *p* values are calculated by one-way ANOVA followed by the Tukey’s test. * *p* < 0.05, ** *p* < 0.01 and *** *p* < 0.001 vs. 0 μM EUP group. EUP: Eupatilin; GSEA: Gene set enrichment analysis; CDK6: Cyclin dependent kinase 6; GAPDH: Glyceraldehyde-3-phosphate dehydrogenase.

**Figure 3 ijms-24-05933-f003:**
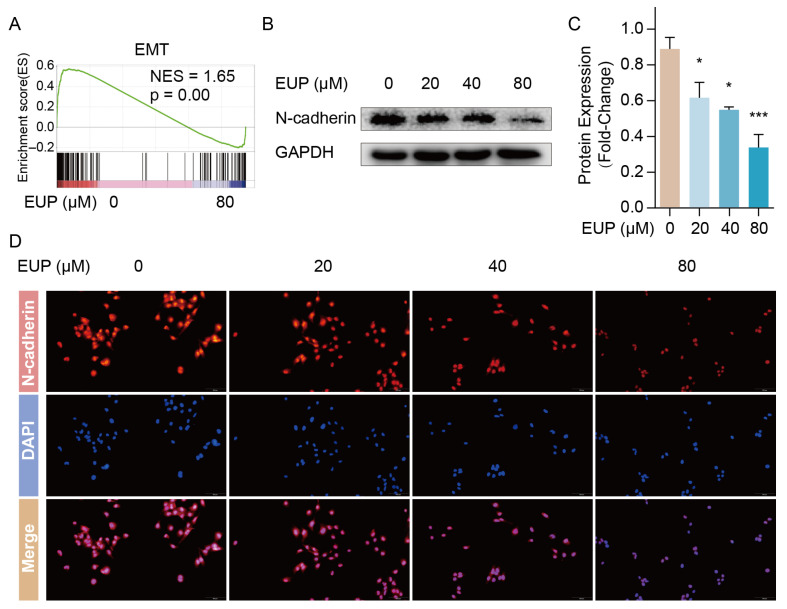
Eupatilin down-regulated genes of EMT process in LX-2 cells. (**A**) Enrichment plot of EMT gene set. (**B**) The expression of N-cadherin protein detected 48 h after eupatilin treatment by Western blotting. (**C**) Relative protein signal intensity was quantified as mean ± standard deviation (mean ± SD, n = 3). *p* values are calculated by one-way ANOVA followed by the Tukey’s test. * *p* < 0.05 and *** *p* < 0.001 vs. 0 μM EUP group. (**D**) Immunofluorescence staining for N-cadherin (red) in LX-2 cells. Nuclei were counter-stained with DAPI (blue). EUP: Eupatilin; EMT: Epithelial–mesenchymal transition; GAPDH: Glyceraldehyde-3-phosphate dehydrogenase; DAPI: 4′,6-diamidino-2-phenylindole.

**Figure 4 ijms-24-05933-f004:**
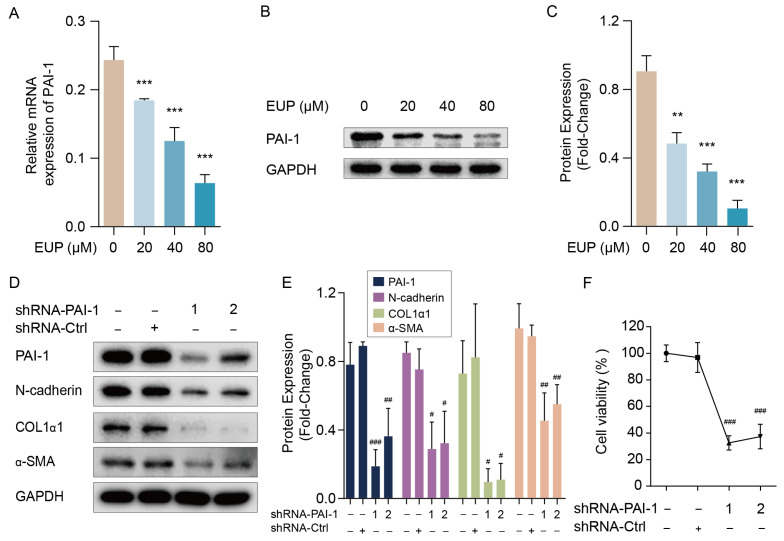
PAI-1 regulates the activation of LX-2 cells involving EMT pathway. (**A**) Relative mRNA expression of PAI-1 in LX-2 cells when treated with eupatilin (mean ± SD, n = 3). (**B**) Western blot analysis of PAI-1 in LX-2 cells. (**C**) Relative protein signal intensity of Figure 4B was quantified as mean ± standard deviation (mean ± SD, n = 3). (**D**) Protein levels of PAI-1, N-cadherin, COL1α1 and α-SMA detected by Western blotting. LX-2 cells were transfected with control (shRNA-Ctrl) or shRNA targeting *PAI-1* (shRNA-PAI-1 1 and shRNA-PAI-1 2) for 48 h. (**E**) Relative protein signal intensity of Figure 4D was quantified as mean ± standard deviation (mean ± SD, n = 3). (**F**) Cell viability of LX-2 cells transfected with shRNA-Ctrl or shRNA-PAI-1 detected by CCK8 assay (mean ± SD, n = 4). *p* values are calculated by one-way ANOVA followed by the Tukey’s test. ** *p* < 0.01 and *** *p* < 0.001 vs. 0 μM EUP group. ^#^
*p* < 0.05, ^##^
*p* < 0.01 and ^###^
*p* < 0.001 vs. shRNA-Ctrl group. EUP: Eupatilin; α-SMA: α-smooth muscle actin; COL1α1: Collagen type I alpha 1; PAI-1: Plasminogen activator inhibitor-1; GAPDH: Glyceraldehyde-3-phosphate dehydrogenase.

**Figure 5 ijms-24-05933-f005:**
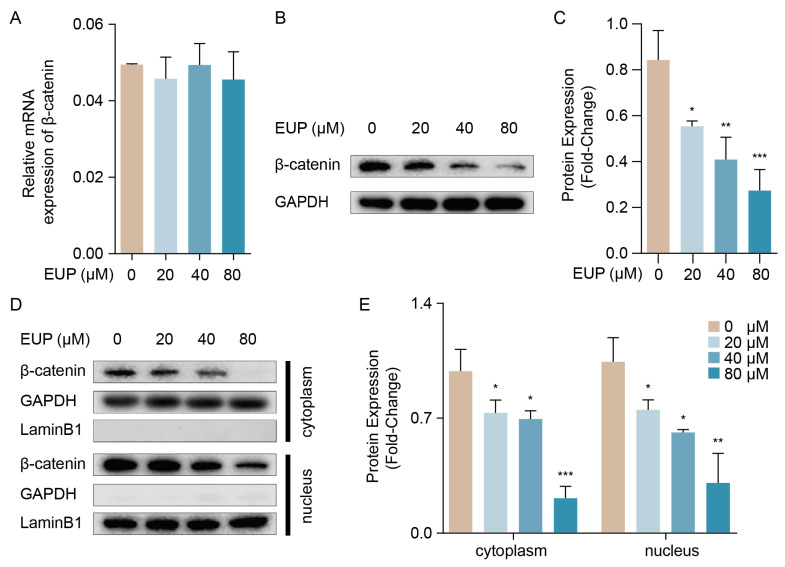
Eupatilin inhibits β-catenin signalling pathway. (**A**) Relative mRNA expression of β-catenin in LX-2 cells treated with eupatilin (mean ± SD, n = 3). (**B**) Western blot analysis of β-catenin in LX-2 cells treated with eupatilin. (**C**) Relative protein signal intensity of Figure 5B was quantified as mean ± standard deviation (mean ± SD, n = 3). (**D**) Western blot analysis of β-catenin protein in both cytoplasm and nucleus. (**E**) Relative protein signal intensity of Figure 5D was quantified as mean ± standard deviation (mean ± SD, n = 3). The mRNA and proteins detection were preformed 48 h after eupatilin treatment. *p* values are calculated by one-way ANOVA followed by the Tukey’s test. * *p* < 0.05, ** *p* < 0.01 and *** *p* < 0.001 vs. 0 μM EUP group. EUP: Eupatilin; GAPDH: Glyceraldehyde-3-phosphate dehydrogenase.

**Figure 6 ijms-24-05933-f006:**
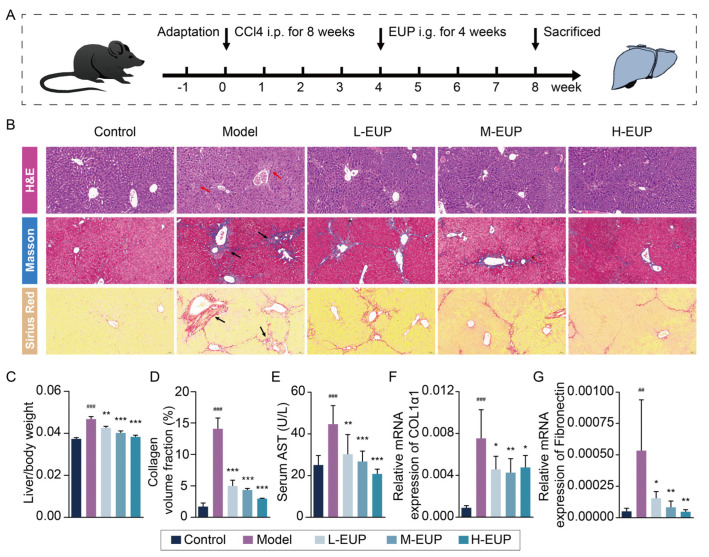
Eupatilin alleviates hepatic fibrosis in CCl4-induced mouse models. (**A**) The scheme of animal experiments in present study. The grouping was as follows: Control; Model (CCl4, without EUP treatment); L-EUP (CCl4, 10 mg/kg EUP); M-EUP (CCl4, 20 mg/kg EUP); H-EUP (CCl4, 40 mg/kg EUP). (**B**) Representative images of livers stained with H&E, Masson, and Sirius red (Red arrows represent mononuclear cell infiltration, black arrows represent collagen deposition. Scale bars = 50 μm). (**C**) The ratios of liver weight to body weight in different groups (mean ± SD, n = 6). (**D**) Analysis of collagen volume fraction of Masson staining. (**E**) Serum levels of AST in CCl4-induced mice (mean ± SD, n = 6). (**F**,**G**) Relative mRNA expression of COL1α1 and Fibronectin in liver tissues (mean ± SD, n = 6). ^##^
*p* < 0.01 and ^###^ *p* < 0.001 vs. control group. * *p* < 0.05, ** *p* < 0.01 and *** *p* < 0.001 vs. model group. *p* values are calculated by one-way ANOVA followed by the Tukey’s test. CCl4: Carbon tetrachloride; i.p.: Intraperitoneal; i.g.: Intragastrical; EUP: Eupatilin; H&E: Hematoxylin and eosin; L-EUP: Low-dose eupatilin; M-EUP: Middle-dose eupatilin; H-EUP: High-dose eupatilin; AST: Aspartate aminotransferase; COL1α1: Collagen type I alpha 1.

**Figure 7 ijms-24-05933-f007:**
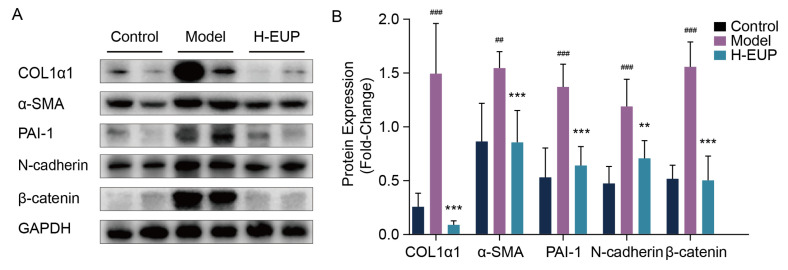
Eupatilin inhibits expression of COL1α1, α-SMA, PAI-1, N-cadherin, and β-catenin in the liver tissues of model group treated with the high-dose eupatilin. (**A**) Western blot analysis of COL1α1, α-SMA, PAI-1, N-cadherin and β-catenin in CCl4-induced mice liver tissues treated with eupatilin (40 mg/kg). (**B**) Relative protein signal intensity was quantified as mean ± standard deviation (mean ± SD, n = 6). ^##^
*p* < 0.01 and ^###^
*p* < 0.001 vs. control group. ** *p* < 0.01 and *** *p* < 0.001 vs. model group. *p* values are calculated by one-way ANOVA followed by the Tukey’s test. H-EUP: High-dose eupatilin; COL1α1: Collagen type I alpha 1; α-SMA: α-smooth muscle actin; PAI-1: Plasminogen activator inhibitor-1; GAPDH: Glyceraldehyde-3-phosphate dehydrogenase; EMT: Epithelial–mesenchymal transition; ECM: Extracellular matrix.

**Figure 8 ijms-24-05933-f008:**
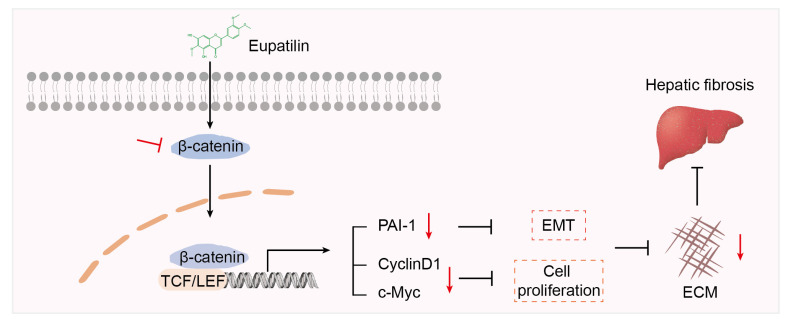
Schematic diagram of the anti-fibrotic effect of eupatilin. Eupatilin inhibits HSC activation by inhibiting EMT process and cell proliferation via suppressing the expression of β-catenin and its downstream targets. Subsequently, inhibition of HSC activation leads to the reduction of the ECM and amelioration of hepatic fibrosis. Red arrows represent decreased expression. Black arrows represent the direction of signal transduction. Red and black blocked lines represent inhibition.

**Table 1 ijms-24-05933-t001:** Primers used in qRT-PCR.

	Gene	Forward primer (5′→3′)	Reverse primer (5′→3′)
Mouse	*GAPDH*	AGGTCGGTGTGAACGGATTTG	TGTAGACCATGTAGTTGAGGTCA
	*COL 1α1*	TAAGGGTCCCCAATGGTGAGA	GGGTCCCTCGACTCCTACAT
	*Fibronectin*	AAAAGTTTGTGGGAGTCGTTCT	GGCCCTGTTCTTCCATCCAG
Human	*GAPDH*	GGCATGGACTGTGGTCATGAG	TGCACCACCAACTGCTTAGC
	*PAI-1*	GCACCACAGACGCGATCTT	ACCTCTGAAAAGTCCACTTGC
	*β-catenin*	CATCTACACAGTTTGATGCTGCT	GCAGTTTTGTCAGTTCAGGGA

## Data Availability

All of the data presented in this study are included in the article, additional supporting materials can be directed to the corresponding authors.

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
