# Peer review of "Eupatilin Ameliorates Hepatic Fibrosis and Hepatic Stellate Cell Activation by Suppressing β-catenin/PAI-1 Pathway"

_ijms, 2023, doi:10.3390/ijms24065933_

Round 1
Reviewer 1 Report
The authors in this manuscript submission present a study that investigated the mechanism of anti-fibrotic properties of bioactive flavone eupatilin using in vitro proliferation of hepatic stellate cell line LX-2 and a CCl4-induced murine model of hepatic fibrosis. The authors suggest that eupatilin significantly repressed the levels of the fibrotic markers COL1α1 and α-SMA as well as other collagens in LX-2 cells and inhibited LX-2 cell proliferation, as indicated by the reduced cell viability and downregulation of c-Myc, cyclinB1, cyclinD1, and CDK6. Moreover, they show that eupatilin decreased the level of PAI-1 in a dose-dependent manner, and knockdown of PAI-1 using PAI-1-specific shRNA significantly suppressed the levels of COL1α1, α-SMA, and the epithelial-mesenchymal transition (EMT) marker N-cadherin in LX-2 cells. Western blot analysis indicated that eupatilin reduced the protein level of β-catenin and its nuclear translocation, while the transcript level of β-catenin was not affected in LX-2 cells. Furthermore, analysis of the histopathological changes in the liver and markers of liver function and fibrosis revealed that hepatic fibrosis in CCl4-treated mice was markedly alleviated by eupatilin. In conclusion, eupatilin ameliorates hepatic fibrosis and hepatic stellate cell activation by suppressing β-catenin/PAI-1 pathway.
The current study does expand upon the previous studies that demonstrated the antioxidant, anti-inflammatory and anti-fibrotic effects of eupatilin and its potential use in the prevention of idiopathic pulmonary fibrosis in vivo (Kim et al., EBioMedicine 2019; Lee et al., Clin Exp Reprod Med 2020; Park et al., PLoS One 2021). In light of these published studies, the novelty of this current study is primarily on the in vivo application of eupatilin to reduce hepatic fibrosis. There are strengths especially in vitro studies involving the proliferation of LX-2 cell lines and treatment conditions (e.g., inhibiting b-catenin/PAI-1 pathway), however, the majority of experiments (total 5 main figures) are focused on LX-2 cell line studies and less focus on in vivo studies (Fig. 6. And part of Fig. 7A). The quantification of hepatic fibrosis from in vivo model can be helpful to interpret the effect of EUP on hepatic fibrosis.
Major Concerns:
1. In Fig. 1, it is not clear at what time points the protein expression of Col1a and a-SMA were measured? Were other proteins also measured (e.g., TGF-b etc.)? Although a dose-dependent effect on Col1a is clearly shown, the effect is subtle on a-SMA (from 40 to 80 mM)? Are bar graphs representative of multiple culture conditions or one (what does error bar represent)? Also, did authors look at a-SMA on transcription level (e.g., heatmap)?
2. In the text of the manuscript (line 95-96) related to Fig. 2, rationale is not clear why authors chose CCK-8 and EdU assays? Same for G2M, E2F, MYC, CyclinB1 and so on? These factors need full description and rationale. Same for EMT in Fig. 3?
3. In Fig. 5, why EUP did not affect the mRNA expression of b-catenin but protein expression decreased? It needs to be discussed.
4. In Fig. 6 (in vivo study), as indicated above, the quantification of hepatic fibrosis could be very helpful (e.g., Ishak scoring for fibrosis) to evaluate the effect of EUP on in vivo model of hepatic fibrosis. The resolution quality of staining especially H&E staining is poor and hard to see the differences among the groups. The authors should use arrows to indicate mononuclear cell infiltration and collagen deposition in liver sections.
5. In Fig. 6, serum AST value in CCl4-treated mice (model) is very low. Likewise, relative mRNA expression of Col1a from different doses do not look convincing (dose-dependent?). Likewise, relative mRNA expression of fibronectin is highly variable in model and not convincing. Did authors measure mRNA expression of a-SMA?
6. The panel A in Fig. 7 should be incorporated with Fig. 6 (or presented separately), the schematic (graphical) representation should be presented by itself in a more informative form.
Minor Concerns:
1. Rationale and the language needs an attention in the manuscript.
2. Figure legends need to be more explanatory and informative.

Author Response
Dear reviewer,
Thank you for your consideration of our manuscript. We appreciate your constructive comments and suggestions, and are very grateful for your positive appraisal of our manuscript. In the revised manuscript, we have attempted to address the concerns raised by the reviewer, and have added the experiments and data requested.
Point 1: In Fig. 1, it is not clear at what time points the protein expression of Col1a and a-SMA were measured? Were other proteins also measured (e.g., TGF-b etc.)? Although a dose-dependent effect on Col1a is clearly shown, the effect is subtle on a-SMA (from 40 to 80 mM)? Are bar graphs representative of multiple culture conditions or one (what does error bar represent)? Also, did authors look at a-SMA on transcription level (e.g., heatmap)?
Response 1: We appreciate the reviewer's comments. In the present study,we prepared the protein samples for detecting COL1α1 and α-SMA expression at 48 hours after eupatilin treatment. And in the revised manuscript, we have added the description of time points for the protein detection in line 88, 124, 142 and 196.
The transcriptome data showed TGFβ2 level were significantly downregulated in eupatilin-treated group. We did not detect the protein level of TGFβ members in present study.
We found that eupatilin significantly inhibited α-SMA protein expression, although it presented a subtle dose-dependent manner. Three independent biological replicates were performed in Western blot assay to ensure the authenticity of the results. And we detected the mRNA level of α-SMA. Unexpectedly, eupatilin did not affect its transcription level. It may be a complex mechanism for α-SMA in response to eupatilin. Up to now, we haven’t found a convincing explanation for this phenomenon. It is an interesting subject to investigate in our future studies.
In present study, three independent biological replicates were performed in Western blot assay. The error bars came from densitometry analysis corresponding to three protein bands.
Point 2: In the text of the manuscript (line 95-96) related to Fig. 2, rationale is not clear why authors chose CCK-8 and EdU assays? Same for G2M, E2F, MYC, CyclinB1 and so on? These factors need full description and rationale. Same for EMT in Fig. 3?
Response 2: We appreciate the reviewer's comments and constructive suggestions. In revised manuscript, we added detailed description for CCK-8 and EdU assays in line 99-108. Partial decription as follows: CCK8 assay is based on measuring the dehydrogenase activity of proliferating cells that are metabolically active and able to transform the slight yellow WST-8 into orange formazan. The CCK8 results are directly proportional to the cell number of viable cells. EdU (5-ethynyl-2’-deoxyuridine) is a nucleoside analog of thymidine and it can incorporate into DNA during DNA synthesis in proliferating cells. CCK8 and EdU assay were often used to measuring cell proliferation.
We added description for selection of G2M, E2F, MYC, CyclinB1 in line 109-114. Partial decription as follows: The results showed that seven gene sets decreased significantly in eupatilin-treated group (P<0.05), and four of them are closely associated with cell proliferation, including G2M checkpoint, E2F targets, MYC targets and mitotic spindle. Subsequently, we selected the key regulator of G2M checkpoint (Cyclin B1), c-Myc and two other crucial regulator of cycle progression (Cyclin D1 and CDK6) to assess the effects of eupatilin on the cell cycle.
We added description for EMT in line 130-134. Partial description as follows: Previous studies demonstrated that EMT was closely associated with hepatic fibrosis, and inhibited EMT process contributes to the suppression of HSC activation. The GSEA analysis shown that the gene set of EMT was significantly downregulated in the eupatilin treated group. Therefore, The EMT, one of seven significant gene sets, caught our attention.
Point 3. In Fig. 5, why EUP did not affect the mRNA expression of b-catenin but protein expression decreased? It needs to be discussed.
Response 3: We appreciate the reviewer's comments. In present study, eupatilin have no effect on the transcriptional level of β-catenin, while the β-catenin protein level was decreased dramatically. We speculated that eupatilin promoted protein degradation of β-catenin or inhibited its translational process. We have added the explanation of this phenomenon in line 276-277.
Point 4. In Fig. 6 (in vivo study), as indicated above, the quantification of hepatic fibrosis could be very helpful (e.g., Ishak scoring for fibrosis) to evaluate the effect of EUP on in vivo model of hepatic fibrosis. The resolution quality of staining especially H&E staining is poor and hard to see the differences among the groups. The authors should use arrows to indicate mononuclear cell infiltration and collagen deposition in liver sections.
Response 4: We appreciate the reviewer's constructive suggestions. We selected the ratio of liver weight to body weight and collagen volume fraction for evaluating the effect of eupatilin on in vivo model of hepatic fibrosis and added in revised Figure 6. In addition, we attemplted to use the standard of Ishak scoring for quantifing the mouse hepatic fibrosis. Unfortunately, it is not easy to apply the criterion which appropriated for human hepatic fibrosis.
We added high resolution staining for H&E staining and used arrows to indicate mononuclear cell infiltration (red arrows) and collagen deposition (black arrows) on liver sections in revised Figure 6.
Point 5. In Fig. 6, serum AST value in CCl4-treated mice (model) is very low. Likewise, relative mRNA expression of Col1a from different doses do not look convincing (dose-dependent?). Likewise, relative mRNA expression of fibronectin is highly variable in model and not convincing. Did authors measure mRNA expression of a-SMA?
Response 5: We appreciate the reviewer's comments. For low AST value, we speculated that storage time of serum sample is the likely cause. The serum sample were stored in ultra-low-temperature refrigerator (-80℃) for about one month before AST assay, because of the maintenance of the microplate reader (BioTek). To assure the reliability of results, we performed the AST assay under the same experimental conditions accroding to the manufacturer’s instructions.
The mRNA level of COL1α1 was significantly decreased in eupatilin-treated groups. Unfortunately, we did not observe the dose dependent in three eupatilin-treated groups. The high variability of model group probably derived from inter-individual heterogeneity. To ensure the reliability of the results, we used six mice for each group in qPCR assay (n=6).
We did not observe significant changes in mRNA expression of α-SMA in eupatilin-treated groups. Up to now, we haven’t found an appropriate explanation for this result.
Point 6. The panel A in Fig. 7 should be incorporated with Fig. 6 (or presented separately), the schematic (graphical) representation should be presented by itself in a more informative form.
Response 6: We appreciate the reviewer's constructive suggestions. We presented the panel A in Figure 7 separately. And the schematic representation was termed as Figure 8 in the revised manuscript.
Minor Concerns:
Point 1: Rationale and the language needs an attention in the manuscript.
Response 1: We appreciate the reviewer's constructive suggestions. In the revised manuscript, we added several rationales to make it easier to read and carefully revised abstracts, texts, charts and references in strict accordance with the requirements of journal, as well as modified the language by native English speaker.
Point 2: Figure legends need to be more explanatory and informative.
Response 2: We appreciate the reviewer's constructive suggestions. We added detailed information in every figure legend accordingly.

Reviewer 2 Report
The purpose of this study is to determine the anti-hepatic fibrosis activity of eupatilin and explore its potential molecular mechanisms. The authors conclude that eupatilin ameliorated hepatic fibrosis and HSC activation by inhibiting β-catenin/PAI-1 pathway. The experimental plan is well designed. The techniques used in this study are applicable and appropriated. The results are clearly presented. However, the context in this article need more detail of the interpretation and discussion.
1. This study investigated the effect of eupatilin on the phenotype of LX2 cell line without any activator. Please describe more about the characteristic of the cell line, why it could be used to represent as hepatic fibrosis, EMT activation and HSC activation models.
2. Treatment of eupatilin causes the decrease of LX2 cell viability. The results show that it could be the anti-proliferative effect of the compound. Is it possible to be its cytotoxicity effect that causes cell death? Moreover, it must be very nice if the author calculate and show %cell viability (relative to non-treated control) of the results from CCK-8 assay.
3. Dose of eupatilin at 40 and 80 uM clearly reduced cell viability which may be not suitable for the experiments which study the mechanism of the compound to inhibit hepatic fibrosis and HSC activation. Especially, when the expression level both mRNA and protein is measured. The concentration used must be non- or sub-toxic dose to avoid the consequence of cytotoxicity or some stress in the cells.
4. PAI knocking down also dramatically decreases the cell number.
As stated in 3&4, Could the authors describe or discuss more how to be sure or confirm that the effect on cell proliferation dose not influent the expression level of N-CAD, COL, SMA, beta-catenin?
5. An mRNA level of catenin is not altered while the protein level is markedly decreased after the treatment of the compound. These may be due to not only the degradation of the protein, but also the inhibition of translation process. The reduction of catenin translocation should be due to the decrease of the protein level, not the direct inactivation of the protein.
6. According to anti-proliferative effect of the compound on LX2 cells, the safety of the compound must be concern. Cytotoxicity testing of the compound on normal hepatocyte or other normal cell type should be investigated.
Moreover, please describe or discuss more about the possible side effects when PAI is down-regulated.
7. In vivo experiment, are there any significant difference of body and other organs weight?
Minor suggestion
- western blot should be revised as Western blot
- Statistical mark represent in each Figure must be described in the figure legend.
Author Response
Dear reviewer,
Thank you for your consideration of our manuscript. We appreciate your constructive comments and suggestions, and are very grateful for your positive appraisal of our manuscript. In the revised manuscript, we have attempted to address the concerns raised by the reviewer, and have added the experiments and data requested.
Point 1: This study investigated the effect of eupatilin on the phenotype of LX2 cell line without any activator. Please describe more about the characteristic of the cell line, why it could be used to represent as hepatic fibrosis, EMT activation and HSC activation models.
Response 1: We appreciate the reviewer's constructive suggestions. In healthy liver, hepatic stellate cells (HSCs) are vitamin A-storing cells in quiescent state. While, liver injury induces quiescent HSCs into activated state. Then the activated HSCs produces a large amount of extracellular matrix (ECM) including collagens and fibronectin. The excessive accumulation of ECM will induce hepatic fibrosis. Therefore, activated HSCs are the key effector cells in hepatic fibrosis. LX-2 is a cell line derived from activated human HSCs, retaining the key features of activated HSCs. Therefore, LX-2 cell line without activator was often used in the experimental studies of hepatic fibrosis. We have detailed characteristic description of LX-2 cells in Introduction section with slight adjustment for easy to read in line 37-38, and Result section in line 74-75 in the revised manuscript.
On the other hand, HSCs undergo a trans-differentiation to mesenchymal phenotype during fibrogenesis. This conversion is considered similar with epithelial-mesenchymal transition (EMT). Therefore, activated HSCs (LX-2) possess the characteristics of the cells in EMT process. We added the description in line 53-56.
Point 2: Treatment of eupatilin causes the decrease of LX2 cell viability. The results show that it could be the anti-proliferative effect of the compound. Is it possible to be its cytotoxicity effect that causes cell death? Moreover, it must be very nice if the author calculate and show %cell viability (relative to non-treated control) of the results from CCK-8 assay.
Response 2: We appreciate the reviewer's comment and constructive suggestions. We observed that the floating-cell numbers in culture (sloughed cells) has no obvious increase with increasing drug concentration of eupatilin. In addition, expression of several proliferation-related genes (c-Myc, CDK6, Cyclin D1 and Cyclin B1) were inhibited in eupatilin-treated groups. So we considered eupatilin exerting an anti-proliferative effect on LX-2 cells. We calculated and shown % cell viability in CCK-8 assay according to your suggestion in the updated Figure 2 and Figure 4.
Point 3. Dose of eupatilin at 40 and 80 uM clearly reduced cell viability which may be not suitable for the experiments which study the mechanism of the compound to inhibit hepatic fibrosis and HSC activation. Especially, when the expression level both mRNA and protein is measured. The concentration used must be non- or sub-toxic dose to avoid the consequence of cytotoxicity or some stress in the cells.
Response 3: We appreciate the reviewer's comment. Activated HSCs are the pivotal effector cells in hepatic fibrosis. Inhibition of cell proliferation ability is a key feature for suppressing HSCs activation. The appropriate concentration of drug that inhibiting HSC proliferation was often selected as exploring the underlying mechanism in numerous studies. Therefore, we selected the dose of eupatilin at 40 and 80 μM in present study. We will adopt your prospective suggestion in our follow-up research.
Point 4. PAI knocking down also dramatically decreases the cell number.
As stated in 3&4, Could the authors describe or discuss more how to be sure or confirm that the effect on cell proliferation dose not influent the expression level of N-CAD, COL, SMA, beta-catenin?
Response 4: We appreciate the reviewer's comment. For PAI-1 knock-down dramatically decreasing the cell number. To avoid the influence of cell proliferation dose on the expression level of proteins, we measured the protein concentration and loaded equal amounts of protein on SDS-PAGE for each condition. In addition, GAPDH was used as a reference to normalize expression levels between samples. We added more description in Materials and Methods section in line 385-387.
Point 5. An mRNA level of catenin is not altered while the protein level is markedly decreased after the treatment of the compound. These may be due to not only the degradation of the protein, but also the inhibition of translation process. The reduction of catenin translocation should be due to the decrease of the protein level, not the direct inactivation of the protein.
Response 5: We appreciate the reviewer’s comments. We fully endorse the reviewer’s opinion that reduction of β-catenin translocation should be due to the decrease of the protein level, not the direct inactivation of the protein. We added the discussion about possible reasons of decreased protein level of β-catenin in eupatilin-treated groups in line 276-277, and revised the schematic diagram.
Point 6. According to anti-proliferative effect of the compound on LX2 cells, the safety of the compound must be concern. Cytotoxicity testing of the compound on normal hepatocyte or other normal cell type should be investigated.
Moreover, please describe or discuss more about the possible side effects when PAI is down-regulated.
Response 6: We appreciate the reviewer’s comments and constructive suggestions. We detected cell cytotoxicity of eupatilin on THLE-3 cells, a normal human liver epithelial cell line. The result indicated that EUP did not significantly influent the THLE-3 cell proliferation.
We described the possible side effects of PAI-1 down-regulation in Discussion section in line 305-311. The discussion is as follows: Numerous studies using models of lung, kidney and liver fibrosis indicated that inhibition of PAI-1 activity or PAI-1 knockout alleviates fibrosis. The present study provided an effective anti-fibrosis drug by inhibiting PAI-1 expression. However, PAI-1 deficiency presented an opposite effect on cardiac fibrosis. Therefore, caution needs to be taken when the treatment of multiple organ fibrosis model by inhibiting PAI-1 expression though eupatilin. In this case, targeted delivery of eupatilin to liver tissue is worthwhile to explore.
Point 7. In vivo experiment, are there any significant difference of body and other organs weight?
Response 7: We appreciate the reviewer’s comments. We analyzed the ratio of liver weight to body weight. When compared with control group, the ratios were significantly increased in the model group. And the ratios were decreased in eupatilin-treated groups in a dose-dependent manner. We added the result description in line 214-216, and updated Figure 6 in revised version.
Minor suggestion
Poin1: western blot should be revised as Western blot
Response 1: We appreciate the reviewer’s comments. We corrected the inconsistent convention of Western blot.
Point 2: Statistical mark represent in each Figure must be described in the figure legend.
Response 2: We appreciate the reviewer’s comments. We added detailed information of statistical mark represent in the updated figure legends.

Round 2
Reviewer 2 Report
The authors well respond to the comments. This revised version can be accepted for the publication.